# Integration of targeted sequencing and pseudo-tetraploid genotyping into clinically assisted decision support for β-thalassemia invasive prenatal diagnosis

Wenguang Jia[1,2,3,4☯], Jiying Shi[1☯], Hengying Zhu[1,2,3,4], Xiaojing Wu[1], Yayun Ling[1], Ping Chen[1,2,3,4]*

1 The First Affiliated Hospital, Guangxi Medical University, Nanning, Guangxi Province, China, 2 NHC Key Laboratory of Thalassemia Medicine, Nanning, Guangxi Province, China, 3 Key Laboratory of Thalassemia Medicine, Chinese Academy of Medical Sciences, Nanning, Guangxi Province, China, 4 Guangxi Key Laboratory of Thalassemia Research, Nanning, Guangxi Province, China

☯ These authors contributed equally to this work.
* chenping@gxmu.edu.cn

**Data Availability Statement:** Genetic data on thalassemia is prohibited from being transmitted to the public database temporarily by Chinese Law,

## Abstract

### Background

The high prevalence of β-thalassemia indicates the severe medical burden in Guangxi province in China. Millions of thousands of prenatal women with healthy or thalassemia-carrying fetuses received an unnecessary prenatal diagnosis. We designed a prospective single-center proof-of-concept study to evaluate the utility of a noninvasive prenatal screening method in the stratification of beta-thalassemia patients before invasive procedures.

### Methods

Next-generation and optimized pseudo-tetraploid genotyping-based methods were utilized in preceding invasive diagnosis stratification to predict the mater-fetus genotype combinations in cell-free DNA, which is from maternal peripheral blood. Populational linkage disequilibrium information with additional neighboring loci to infer the possible fetal genotype. The concordance of the pseudo-tetraploid genotyping with the gold standard invasive molecular diagnosis was used to evaluate the effectiveness of this method.

### Results

127 β-thalassemia carrier parents were consecutively recruited. The total genotype concordance rate is 95.71%. The Kappa value was 0.8248 for genotype combinations and 0.9118 for individual alleles.

### Conclusion

This study offers a new approach to picking out the health or carrier fetus before invasive procedures. It provides valuable novel insight into patient stratification management on β-thalassemia prenatal diagnosis.

but it can be disclosed in a restricted way. If necessary, please contact to Ethics Committee of the First Affiliated Hospital of Guangxi Medical University (Tel: +867715356557, Fax: +867715359801, Address: No.6, Shuangyong Road, Qingxiu District, Nanning, Guangxi province, 530021, China. We also provide a point of contact through e-mail which can contact the Ethics Committee of the First Affiliated Hospital of Guangxi Medical University, and request access to our data: gxydkyb@163.com.

**Funding:** This study was funded by National Nature Science Foundation of China (81960574, 82060578) and the Non-profit Central Research Institute Fund of Chinese Academy of Medical Sciences (2019PT310012).

**Competing interests:** The authors have declared that no competing interests exist.

## Introduction

Beta thalassemia (β-thalassemia) is amongst the commonest single-gene disorders worldwide, with the highest prevalence in the Mediterranean region, southeast Asia, sub-Saharan Africa, and the Middle East [1–3]. Over 80 million people (1.5%) worldwide may be carriers of β-thalassemia, and it is estimated that 60,000 affected infants are born yearly [4, 5]. At the same time, only limited prenatal diagnosis choice has been offered for clinical application [3]. In China, thalassemia is mainly prevalent in southern China, south of the Yangtze River. The population carrying frequency (combined α-/β-/α+β- thalassemia) 31.92% [6], 24.19% [7], 23.12% [8], 19.48% [9], 6.8% [10], and 2.63% [11] in the top six provinces Yunnan, Guangxi, Hainan, Guangdong, Fujian, and Sichuan, respectively. A recent meta-analysis suggests that the overall prevalence of α-thalassemia, β-thalassemia, and α+β-thalassemia was 7.88%, 2.21%, and 0.48% [12], respectively, in China, and the high prevalence emerged as a significant impact on public health, society, and economy.

Mutations, including single-nucleotide substitutions or small insertions and deletions within *HBB* or the immediate flanking sequence, lead to the reduction or complete abolishment of expression of functional *HBB* chains and cause β-thalassemia [13]. *HBB*-like globin cluster, located on the short arm of chromosome 11 in a late replicating region, methylated with a relatively low number of genes, which are maintained as closed-chromatin (heterochromatin) in non-erythroid cells [14, 15]. β-thalassemia major causes severe anemia, which becomes apparent 3–6 months after birth, when the switch from fetal globin to adult is almost completed [16, 17].

Currently, there is a lacking of effective treatments for thalassemia. Thalassemia major patients require regular blood transfusions, which cause organ iron overload, particularly in the heart, liver, and other endocrine glands, impairing their function [1, 18]. Detection of carriers and subsequent prenatal screening and diagnosis of fetal thalassemia, as well as early intervention in critically ill patients, are essential and consistent with the focus on congenital disabilities prevention. Thus, prenatal testing of fetuses is the most direct and effective way to control the birth of children with severe thalassemia. In China, Guangxi province has the highest incidence of thalassemia. The incidence of α-, β-, and α+β-thalassemia in Guangxi is 12.51%-15.35%, 5.11%-6.64%, and 1.47%-2.08%, respectively [19, 20], and the newly-born population in 2019 is 660,000 (http://tjj.gxzf.gov.cn//tjsj/tjnj/material/tjnj20200415/2020/zk/indexch.htm). Considering the α-/β-thalassemia combined actual incidence, more than 100,000 pregnant women a year require prenatal diagnosis of fetal thalassemia in this province alone, which takes up a lot of medical resources and places a heavy burden on the local medical system.

Despite the high prevalence and importance of prenatal screening for β-thalassemia, there remains a particular paucity of optional approaches for clinical application. The routine clinical diagnosis of thalassemia is to use invasive prenatal diagnostic techniques such as amniocentesis, chorionic villus sampling, or umbilical cord puncture to obtain fetal cells and perform thalassemia gene detection. It inevitably carries a risk of infection and a fetal loss rate of approximately 1% [21–24]. However, even in cases where both parents are carriers of the same recessive monogenic disorder, the probability that the child is homozygous is only one in four, meaning that 75 percent of diagnoses result in a healthy child or a carrier who does not require further treatment. In practice, the precious fetus for infertile couples reduces their courage to bear the risk of an interventional prenatal diagnosis. Therefore, prenatal diagnosis of thalassemia in high-incidence areas is in a dilemma. Families and fetuses are at high risk of disease without the invasive prenatal diagnosis. In contrast, timely diagnosis requires tremendous medical resources and facing patient communication pressure for safety and excessive

diagnosis. More recently, Noninvasive prenatal testing of common chromosome abnormality has rapidly been incorporated into routine clinical practice over the past decade, showing the potential to revolutionize the management of pregnancies at risk of genetic disorders [25]. Previous studies have also demonstrated the potential of a combined target region capture sequencing approach and pseudo-tetraploid genotype analysis for single-gene prenatal detection [26, 27].

Given this, this study aims to develop an invasive prenatal diagnosis decision-making aid method for parents that are β-thalassemia carriers. In this way, most of the healthy or carrier fetuses can be screened out and excluded from additional invasive surgery, ensuring the homozygous patients have access to the invasive medical diagnostic valuable resource. We intend to optimize existing bioinformatics algorithms to improve sensitivity and specificity further to effectively provide an attainable clinical technique for assessing the need for prenatal diagnosis of β-thalassemia and reducing unnecessary medical diagnoses.

## Materials and methods

### Ethical approval

The ethics committee approved the research from The First Affiliated Hospital of Guangxi Medical University, Guangxi, China (No. 2019-KY-NFSC-238). Written informed consent for enrollment or consent to continue and to use patient data was obtained from either the patient or the patient's closest relatives. All procedures in studies involving human participants were performed in compliance with the ethical guidelines of the Helsinki Declaration. Authors had access to get information on patients that could identify individual participants during or after data collection.

### Sample collection and sequencing

As a prospective single-center proof-of-concept study, we consecutively recruited singleton pregnant women preparing for amniocentesis, chorionic villus sampling, or cordocentesis from April 1 to December 31, 2020, at the First Affiliated Hospital of Guangxi Medical University in Guangxi Province, which the highest prevalence of β-thalassemia in China. All included patients had a diagnostic confirmation of at least one of the biological parents of the fetus for β-thalassemia carriers included the following types: *HBB*: CD 41/42(-TCTT), *HBB*: CD 17 (A>T), *HBB*: IVS-II-654(C>T), *HBB*: 26(G>A, Hb E), CD 71/72(+A), *HBB*: -28(A>G), *HBB*: -29(A>G), *HBB*: CD 43(G>T), *HBB*: IVS-I-1(G>T), *HBB*: CD27/28M (+C), *HBB*: CD130M (T>A). Patients with vanished twins, malignancy, autoimmune disease, or other factors that are not appropriate for noninvasive prenatal testing were excluded. There is no direct blood relation and collateral blood relation between up to three generations of the couple.

There is 5 ml of maternal peripheral blood samples collected in Streck tubes onsite from consenting participants. The centrifuging, and the preparation of maternal DNA, cell-free DNA (cfDNA) isolation, library construction, barcoding, sequencing, and data analysis were performed according to previously published methods [26, 27]. Furthermore, 7-bp unique molecular identifier (UMI) was used to determine the real, original, single-allelic molecules [28]. For each maternal plasma sample, an average of 7.7 million reads with 40 bp in length and Q30 > 95% was generated for further analysis.

All fetal genotypes were subjected to standard invasive prenatal diagnosis according to standard protocols for molecular diagnostic testing [29, 30] at the Guangxi Key Thalassemia Research Laboratory. Chorionic villus or amniotic fluid samples of the fetuses were collected by ultrasound-guided transabdominal chorionic villus sampling or amniocentesis, and then DNA was extracted for PCR. Following that, reverse dot blot hybridization was performed,

with 24 oligonucleotide probes arranged on the hybridization membrane to detect seven common and ten rare sites of β-thalassemia mutations in the Chinese population.

## Design principle

Compared to the previous attempt to directly target mutational hotspots [26, 27], in this work, we explore the direction of using populational linkage disequilibrium information with additional neighboring loci to infer the possible fetal genotype of pathogenic locus from targeted sequencing of maternal plasma. For those listed known hotspots of β-thalassemia, we evaluate linkage information from common polymorphic loci from the 1000 Genome project, focusing on EAS individuals. We employ the following search scheme:

1. For each known polymorphic loci as the center, flanking regions of 50 bp on each side were considered candidate target probe regions, and their GC content, uniqueness, and palindromic status were evaluated according to the previous study [28].

2. For each hotspot, the closest 30 loci upstream and downstream are covered by the continuous span of the LD block constructed from the 1000 Genome data.

## Analytical pipeline

gDNA samples from the collection were processed using the same targeting protocol. Sequencing data were aligned to the targeted regions of the human reference genome (hg19). Genomic variants were called using GATK [29]. Haplotyping for each gDNA sample library was processed using Shape it [30]. Pathogenic haplotypes of the known parental carrier were selected and annotated as H1, while the normal haplotypes were grouped as H0.

cfDNA data were pre-processed similarly as previously described [26, 27], with maternal-fetal genotype combinations estimated using pseudo-tetraploid genotyping (PTG). As a follow-up step, fetal genotypes of flanking loci for each hotspot were evaluated against the two haplotype sets formed using the known gDNA data. The fetal cell-free DNA concentration should exceed the detection range (4%) by evaluating the difference between fetal and maternal loci [26]. A likelihood function is defined as the following, and the stepwise accumulated likelihood of both H1 and H0 could be visualized gradually as a function of checked loci within the haplotype.

$$LLK_{s,i} = \prod_{H_i\{1,\ldots,s\}} P(G_{is}|H_{is})$$

Fixme, Select the top-scoring haplotype pair for the test sample and check if it matches.

Fixme, needs to combine the two results somehow, one from PTG, and one from the haplotyping LLK.

## Statistical analysis

All data from this study were inputted into SPSS statistics 22 software (IBM, USA) for statistical analysis.

Clinical data such as pregnancy age, gestation week, weight, height, body mass index (BMI), and other measurements were processed based on numerical distribution. The ones corresponding to a normal distribution were described statistically with a standard deviation of the mean. For data with skewed distribution, the median was used for statistical description.

The concordance of the pseudo-tetraploid genotyping with the gold standard invasive molecular diagnosis was used to evaluate the effectiveness of this method. The sensitivity,

**Table 1. Clinical information of enrolled subjects (pregnant women, n = 127).**

| | Mean±standard deviation | minimum | Maximum |
|---|---|---|---|
| Age (years) | 28.47±6.05 | 16 | 47 |
| Height (cm) | 156.44±5.02 | 145 | 171 |
| Body weight (kg) | 54.78±7.84 | 40.5 | 79.0 |
| Gestational age (weeks) | 15.24±3.31 | 12.0 | 26.0 |
| BMI(m$^2$/kg) | 20.3±3.2 | 18.4 | 23.98 |
| | Fetuses with thalassemia major | Fetuses with thalassemia minor | Healthy fetuses |
| Reproductive history | 2 | 21 | 104 |
| Family history | 3 | 43 | 81 |

cm, centimeter; kg, kilogram; BMI, body mass index

specificity, and Kappa values for genotype combinations and individual alleles were calculated, respectively.

## Results

A total of 127 singleton pregnant women from Guangxi provinces in China were prospectively recruited, and these patients all completed the trial and were included in the statistics. Clinical data are shown in Table 1. The β-thalassemia carrier status of fetal parents is presented in Table 2. *HBB*: CD 41/42 (-TCTT) and *HBB*: CD 17 (A>T) are the mainstream types of β-thalassemia carriers in this continuous sampling study in Guangxi province, accounting for 43.80% (106 in 242) and 32.64% (79 in 242) respectively. Eighty-nine pregnant women in the study had amniocentesis, 37 had chorionic villus sampling, and only one had cordocentesis. Gold standard invasive molecular diagnosis of 127 fetuses revealed seven mutant homozygous and 19 composite heterozygous fetuses with severe thalassemia in both cases. The remaining fetuses were 64 carriers and 37 healthy fetuses.

Compared with the original genotyping approach [26, 27], our new populational linkage disequilibrium information enhanced pseudo-tetraploid genotyping method increases the mutation ratio bias elimination, which is fundamental in genotype calling. The skew allelic

**Table 2. The β-thalassemia carrier status of fetal biological parents.**

| Mutant type | HGVS name | Carrier | |
|---|---|---|---|
| | | Mother | Father |
| CD 41/42 (-TCTT) | *HBB*:c.126_129delCTTT | 56 | 50 |
| CD 17 (A>T) | *HBB*:c.52A>T | 41 | 38 |
| -28 (A>G) | *HBB*:c.-78A>G | 9 | 7 |
| IVS-II-654 (C>T) | *HBB*:c.316-197C>T | 8 | 6 |
| 26 (G>A, Hb E) | *HBB*:c.79G>A | 3 | 7 |
| CD 71/72 (+A) | *HBB*:c.216_217insA | 2 | 4 |
| IVS-I-1 (G>T) | *HBB*:c.92+1G>T | 4 | 1 |
| CD 43 (G>T) | *HBB*:c.130G>T | 1 | 2 |
| CD27/28M (+C) | *HBB*:c.84_85insC | 1 | 0 |
| -29 (A>G) | *HBB*:c.-79A>G | 0 | 1 |
| CD130M (T>A) | *HBB*:c.393T>A | 1 | 0 |
| Total | | 126 | 116 |

*HGVS: Human Genome Variation Society

ratios are mainly caused by the noise from the homologous sequences with the target sequences and also can be generated by random sequencing errors, non-uniformity of coverage, or unbalanced PCR amplification efficiency between maternal and fetal cfDNA. In genotyping methods where pseudo-tetraploid analysis is performed only at the target allele locus, the noise and random error cannot be rechecked and eliminated. On the contrary, adding upstream and downstream linkage sites for joint inference of the possible mother-fetal joint genotype of cfDNA compound in maternal peripheral blood makes the allele ratio closer to the true value. Thus, the predicted genotype would be closer to the true genotype and could be more easily distinguished from other false genotypes (Fig 1).

The cfDNA in maternal peripheral blood is a mixture of the mother and fetus so it can be regarded as a pseudo-tetraploid genotype. This study detected 11 common β-thalassemia mutation sites in each of 127 enrolled pregnant women, resulting in 1397 genotype combinations. Suppose we define the wild-type and mutant alleles of the mother as the letters 'A' and 'B' and the wild-type and mutant alleles of the fetus as 'a' and 'b'. In that case, there are five pseudo-tetraploid genotype combinations in the maternal cfDNA when the couple is a β-thalassemia carrier. Thus, there are 1337 concordant cases among 1397 detected phenotypes with a total genotype concordance rate of 95.71% (Table 3). Furthermore, homozygous and complex heterozygous mutated alleles need to be diagnosed for the fetus. In contrast, wild-type alleles and carriers are theoretically not life-threatening and may be protected from further treatment. Thus, we compared the concordance of our predictions with gold standard data on the necessity of invasive prenatal diagnosis in 1397 genotype combinations, and the Kappa value was 0.8248 (Table 4). To be more precise, each gene locus has two alleles. The detection effectiveness of this method for individual alleles is shown in Table 5, and the final sensitivity was 82.88%, the specificity was 98.84%, and the Kappa value was 0.9118.

In addition, in the current clinical treatment pathway, 118 cases with both parents being carriers in 127 patients require an invasive diagnosis. However, the invasive molecular diagnosis result shows only five fetuses that are pathology homozygous, 17 are complex heterozygous, and 96 people, including 34 healthy individuals and 62 β-thalassemia carriers, underwent unnecessary invasive diagnoses. On the contrary, our method allows for a tremendous reduction of the invasive demand, and only 30 people (4 homozygous patients, 12 compound heterozygotes, and 14 prediction errors) need an additional diagnosis. However, we still missed one pathogenic homozygous and five complex heterozygous children who required invasive procedures.

## Discussion

In China, prenatal management of the thalassemia burden remains a challenging issue due to the sheer size of the population and the high carrier rates in individual provinces. With limited options, prenatal diagnosis of thalassemia relies primarily on invasive procedures associated with risks for both the fetus and the mother. Here, we optimized the existing method based on targeted sequencing and pseudo-tetraploid genotyping and established the first invasive diagnosis stratification and fetus β-thalassemia risk prediction model.

Severe β-thalassemia requires a lifelong dependence on blood transfusions and iron removal, which are expensive and have an extremely high mortality rate, placing a huge burden on patients, families, and society [17]. β-thalassemia is a serious public health problem in endemic regions. Carrier screening and genetic counseling for populations in endemic areas, prenatal fetal genetic testing in high-risk couples, and pregnancy intervention to avoid the birth of children with severe β-thalassemia are the only effective measures for preventing and controlling thalassemia [1]. Publicity and education are essential for high-risk screening

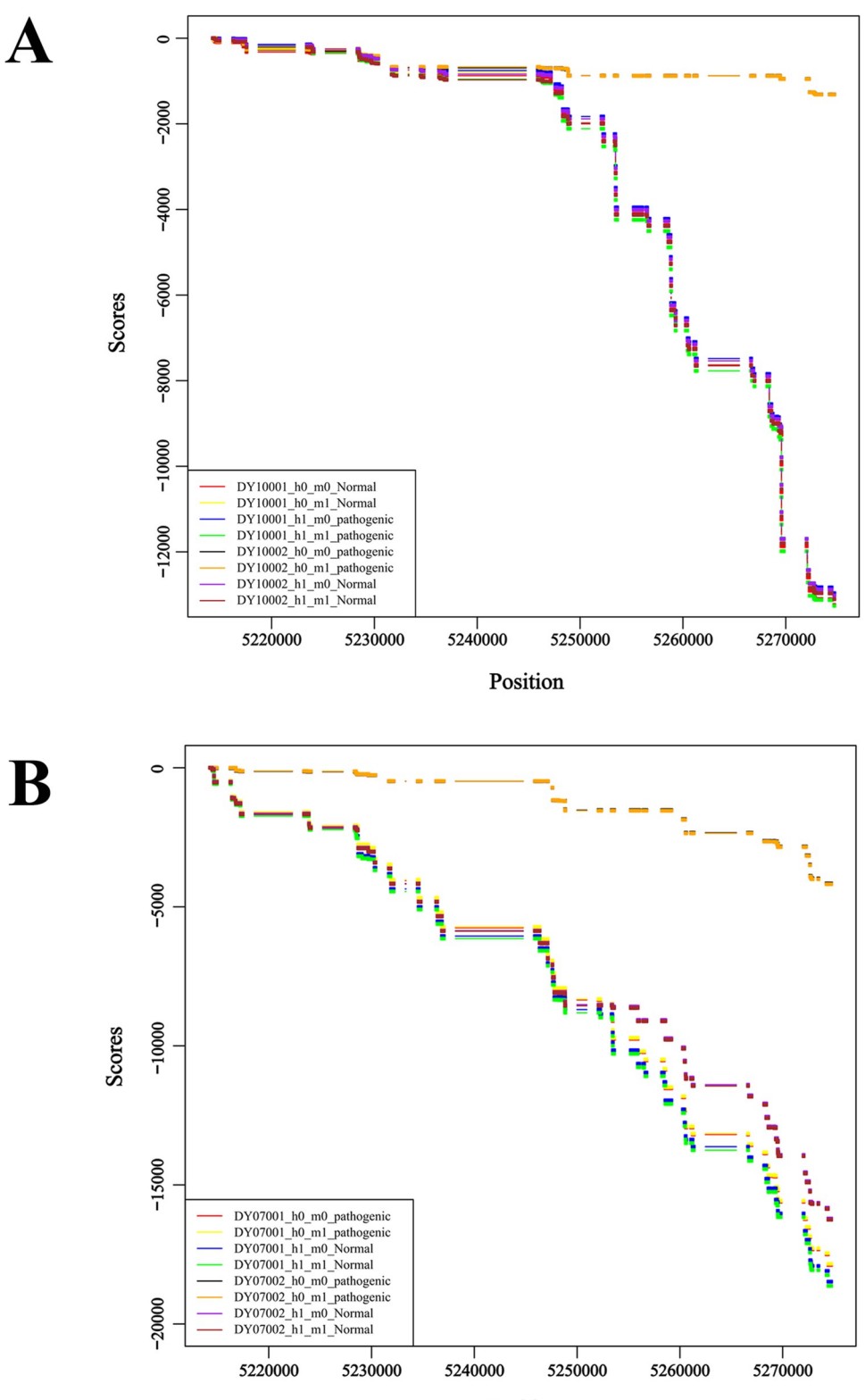

**Fig 1. Diagram of pseudo-tetraploid genotype prediction using the original and optimized methods in sample DY07001.** The pathogenic haplotypes of the known parental carrier were annotated as H1, while the normal haplotypes were grouped as H0. The pathogenic haplotypes of the offspring were marked as M1 and the healthy haplotype as M0. (A) The most likely combination of maternal-fetal genotypes (orange and black) predicted using the

original method among the eight genotypes of a pair of alleles. (B) The most likely combination of maternal-fetal genotypes (orange and black) was predicted with the optimized method. The accurate combination of maternal-fetal genotypes could be more clearly distinguished from other interfering genotypes after modification.

couples. Two of the 127 couples who participated in this study had previously had anemic children, and more than a quarter of the subjects did not consider themselves anemic before this screening. This status underscores the importance of scientific advocacy for prenatal diagnosis in regions where thalassemia is endemic.

The complexity of the spectrum of the β-thalassemia gene mutation complicates β-thalassemia screening and prenatal diagnosis in high-risk couples. More than 200 types of β-thalassemia gene mutations have been found at home and abroad, covering the promoter region, exon region, cut region, and 3' untranslated region of the *HBB* gene. Epidemiological studies have demonstrated that there are significant geographical and ethnic differences in the types of gene mutations [7, 31, 32]. For example, *HBB*:c.79G>A carries 50% in India in South Asia, whereas it carries 0.1% in Pakistan [33]. A total of 46 β-thalassemia mutations have been reported in the Chinese populations. Eight types, including *HBB*:c.126_129delCTTT, *HBB*:c.316-197C>T, *HBB*: c.216_217insA, *HBB*: c.52A>T, *HBB*: c.-78A>G, *HBB*: c.79G>A, *HBB*: c.-79A>G, *HBB*: c.130G>T accounted for more than 95% of the total number of β-thalassemia [7, 31, 34]. On this basis, our group designed a probe capture panel with 12 gene mutation types, including the eight mutations mentioned above, to meet the epidemiological and clinical needs. This study is also the study with the largest number of mutation types in noninvasive prenatal exploration.

Obtaining detectable fetal DNA is the key to the success or failure of prenatal diagnosis. Invasive prenatal diagnosis can be performed by chorionic villus sampling in the first trimester and amniocentesis or cordocentesis in the second trimester [35, 36]. These puncture techniques have become the standard methods for obtaining fetal tissue, which can often extract sufficient fetal DNA to meet the needs of conventional thalassemia genetic detection methods, such as multiplex PCR, nested PCR, and digital PCR. They do not require complex bioinformatics analysis to make accurate judgments. However, these procedures lead to unavoidable risks, such as 2% to 3% fetal loss and 1% placental abruption [23, 37, 38], prompting and driving the continuous exploration of noninvasive prenatal testing for thalassemia.

Cell-free fetal DNA in maternal peripheral blood is an ideal method for noninvasive prenatal detection of genetic material in thalassemia fetuses. In 1997, Lo YM et al. first found the presence of cfDNA in maternal peripheral plasma [39], and they described the biological characteristics of cfDNA, such as concentration, origin, fragment length, and half-life, in a subsequent study [40]. In 2010, it was found that cfDNA contains all the information of the fetal genome [41]. Since then, multiple teams have explored using fetal genetic material in cfDNA

**Table 3. Maternal-fetal genotype combinations concordance of the pseudo-tetraploid genotyping (PTG) with the gold standard invasive molecular diagnosis (IMD).**

| | | IMD genotype | | | | |
|---|---|---|---|---|---|---|
| | | AAaa | AAab | ABaa | ABab | ABbb |
| PTG genotype | AAaa | 1232 | 6 | 5 | 3 | |
| | AAab | 1 | 31 | | 1 | |
| | ABaa | | 1 | 44 | 11 | |
| | ABab | | | 11 | 27 | 1 |
| | ABbb | | | 1 | 19 | 3 |
| Total (1397) | | 1233 | 38 | 61 | 61 | 4 |

**Table 4. The concordance of pseudo-tetraploid genotyping (PTG) with the invasive molecular diagnosis (IMD) (the current gold standard) on the necessity of invasive prenatal diagnosis.**

| PTG | IMD | | |
|---|---|---|---|
| | | Diagnosis inevitable | Relatively safe |
| | Diagnosis inevitable | 36 | 18 |
| | Relatively safe | 6 | 1337 |
| Sensitivity | 85.71% | | |
| Specificity | 98.67% | | |
| PPV | 66.67% | | |
| NPV | 99.55% | | |
| Kappa | 0.8248 | | |

Homozygous and complex heterozygous mutated alleles need to be diagnosed, while wild-type alleles and carriers are theoretically not life-threatening and may be protected from further treatment.

**Table 5. The concordance of pseudo-tetraploid genotyping (PTG) with the invasive molecular diagnosis (IMD) (the current gold standard) on the β-thalassemia individual allele.**

| Mutation ID | PTG | IMD | | Sensitivity | Specificity | PPV | NPV | $P_0$ | $P_e$ | Kappa |
|---|---|---|---|---|---|---|---|---|---|---|
| | | b | a | | | | | | | |
| *HBB*:c.126_129delCTTT | b | 41 | 20 | 93.18% | 90.48% | 67.21% | 98.45% | 0.9094 | 0.6748 | 0.7215 |
| | a | 3 | 190 | | | | | | | |
| *HBB*:c.52A>T | b | 31 | 7 | 73.81% | 96.70% | 81.58% | 94.91% | 0.9291 | 0.7401 | 0.7273 |
| | a | 11 | 205 | | | | | | | |
| *HBB*:c.-78A>G | b | 4 | 2 | 66.67% | 99.19% | 66.67% | 99.19% | 0.9843 | 0.9614 | 0.5916 |
| | a | 2 | 246 | | | | | | | |
| *HBB*:c.316-197C>T | b | 4 | 1 | 80.00% | 99.60% | 80.00% | 99.60% | 0.9921 | 0.9690 | 0.7457 |
| | a | 1 | 248 | | | | | | | |
| *HBB*:c.79G>A | b | 4 | 1 | 100.00% | 99.60% | 80.00% | 100.00% | 0.9961 | 0.9728 | 0.8550 |
| | a | 0 | 249 | | | | | | | |
| *HBB*:c.216_217insA | b | 4 | 0 | 100% | 100% | 100.00% | 100.00% | 1.0000 | 0.9767 | 1.0000 |
| | a | 0 | 250 | | | | | | | |
| *HBB*:c.92+1G>T | b | 1 | 0 | 33.33% | 100.00% | 100.00% | 99.21% | 0.9921 | 0.9922 | -0.0039 |
| | a | 2 | 251 | | | | | | | |
| *HBB*:c.130G>T | b | 1 | 0 | 100.00% | 100.00% | 100.00% | 100.00% | 1.0000 | 1.0000 | 1.0000 |
| | a | 0 | 253 | | | | | | | |
| *HBB*:c.84_85insC | b | 1 | 0 | 100.00% | 100.00% | 100.00% | 100.00% | 1.0000 | 1.0000 | 1.0000 |
| | a | 0 | 253 | | | | | | | |
| *HBB*:c.-79A>G | b | 1 | 0 | 100.00% | 100.00% | 100.00% | 100.00% | 1.0000 | 1.0000 | 1.0000 |
| | a | 0 | 253 | | | | | | | |
| *HBB*:c.393T>A | b | 0 | 0 | NA | 1 | NA | 1.0000 | 1.0000 | 1.0079 | 1.0000 |
| | a | 0 | 254 | | | | | | | |
| All | b | 92 | 31 | 82.88% | 98.84% | 74.80% | 99.29% | 10.8030 | 112.1700 | 0.9118 |
| | a | 19 | 2398 | | | | | | | |

The letter a denote fetal wild-type alleles and b represents mutated alleles.

* NA: not available.

for noninvasive prenatal testing for β-thalassemia. These studies can be divided into two broad categories from the perspective of DNA amplification techniques: quantitative PCR and high-throughput sequencing. The sensitivity and specificity of droplet digital PCR combined with relative variation dosage method for prenatal detection of *HBB*: c.93-21G>A (IVSI-110 G>A) mutation [42] were 100% and 97.07%, respectively. A total of 36 fetuses with homozygous mutations were detected by real-time PCR allele-specific amplification refractory mutation system, and its specificity is consistent with invasive testing [43]. These studies have the shortcomings of a few gene mutations and few experimental samples, which limit their promotion and application.

Maternal cell-free DNA background in maternal peripheral blood is an important factor affecting noninvasive prenatal testing, especially when both husband and wife carry the same β-thalassemia gene mutation [44]. Therefore, the analysis and processing methods of sequencing products after PCR sequencing are the key to its application in noninvasive prenatal testing. Researchers have designed a variety of bioinformatics analysis research protocols to attempt to solve the above problems. Among them, the more commonly used algorithm models are relative haplotype dosage (RHDO) [45, 46] and relative mutation dosage (RMD) [4, 47, 48]. Although some strategies have reached 99.19% sensitivity and 99.92% specificity [4], the current RHDO and RMD strategies still cannot meet the requirements of clinical prenatal testing applications. The main reasons that affect the sensitivity and specificity of the RHDO strategy are the identification of a large amount of parental genetic information and the processing method of the following information when constructing linkage disequilibrium. The main factors affecting the application of RMD are the number of loci in a panel, the number of samples that can be detected in one experiment, and the accurate quantification of wild-type and mutant copy numbers of loci, which are difficult to achieve clinically usable levels at the same time. In contrast, our new method only uses maternal blood to obtain the predictive value of fetal genotypes without recruiting parents and adding additional parental experiments like the RHDO method. In addition, the application of high-throughput sequencing makes it possible to detect multiple sites in multiple patients simultaneously, which will help reduce the cost of clinical use.

This study optimized the existing methods based on targeted sequencing and pseudo-tetra-ploid genotyping and, for the first time, established a predictive model for invasive diagnostic stratification and fetal β-thalassemia risk prediction. The new haplotype-based inference method is improved compared to the previous method. The Kappa value of the single allele was 0.9118, and the Kappa value of the genotype combination was 0.8248, which could be used as a choice for screening healthy or carrier fetuses before invasive diagnosis. Therefore, the results of this pilot study will be used to prevent unnecessary treatment of 82 low-risk patients. However, stratification of some patients for β-thalassemia according to existing methods is ineffective. One pathogenic homozygote and five compound heterozygotes requiring invasive diagnosis were missed, and 14 prediction errors resulted in additional amniocentesis diagnoses. As reported in previous studies, the detection rate and accuracy of the seven genotypes of cfDNA in maternal peripheral blood are variable [25]. Since this method assumes a strong 1:1 equilibrium of allele coverage at heterozygous loci, the significant deviation from the minor allele frequency of 0.5 is contributed by the homozygous genotype at the same locus from the fetus [25]. It means that if the mother is a heterozygous carrier, it is difficult to differentiate the carrier fetus (ABAB) from the affected fetus (ABBB). Such unexpected deviation from equilibrium is relatively common in next-generation sequencing and is responsible for most fetal genotype prediction errors in this study. In addition, the new method still suffers from the fetal genotyping error from allele imbalance, UMI, and a more linear amplification method to the rescue, which is the goal to strive for in the future. In addition, the limited number of

patients may restrict the validity of the background population database, and a larger population background is still needed. The direct haplotyping methods used to improve background quality are as important as improving accuracy. Moreover, China is a multi-ethnic country, and the number of ethnic groups in Guangxi is particularly rich, and the type of β-thalassemia is diverse. Since the above factors may lead to changes in linkage disequilibrium information in the population, our results must be confirmed in other populations, and other β-thalassemia types before large-scale clinical trials as the populational linkage disequilibrium information may change due to those factors mentioned above.

In conclusion, the present study provides valuable new insights into patient stratification before invasive procedures for prenatal diagnosis of β-thalassemia. The predictive ability of cfDNA and noninvasive technologies based on next-generation sequencing will bring better solutions to medical challenges. However, prenatal diagnosis is the only effective way to avoid the birth of the affected fetus. If both parents are present, direct testing of the parents using genomic DNA to predict the risk of a pregnancy giving birth to a child with β-thalassemia would be a more effective, cost-effective, and specific approach. Our strategy is intended only as a reference for clinicians and not as a replacement for invasive testing. Our implementation aims to explore and refine this strategy with a view to its application in clinical practice soon. After all, safe and effective noninvasive prenatal diagnostic techniques have always been the direction and goal of prenatal diagnosis of genetic diseases such as thalassemia.

## Acknowledgments

We thank the patients for participating in this study. We also thank Annoroad Gene Technology Co., Ltd for its technical support and advice in bioinformatics analysis.

## Author Contributions

**Conceptualization:** Wenguang Jia, Ping Chen.

**Data curation:** Wenguang Jia, Jiying Shi, Hengying Zhu.

**Formal analysis:** Wenguang Jia, Jiying Shi, Hengying Zhu.

**Funding acquisition:** Wenguang Jia, Ping Chen.

**Investigation:** Wenguang Jia, Jiying Shi.

**Methodology:** Wenguang Jia, Ping Chen.

**Project administration:** Wenguang Jia, Ping Chen.

**Supervision:** Jiying Shi.

**Validation:** Xiaojing Wu, Yayun Ling.

**Visualization:** Wenguang Jia, Jiying Shi.

**Writing – original draft:** Wenguang Jia, Jiying Shi.

**Writing – review & editing:** Xiaojing Wu, Yayun Ling.

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
