## [Decision Letter · Decision Letter 0]

8 Nov 2022

PONE-D-22-16457Integration of targeted sequencing and pseudo-tetraploid genotyping into clinical assisted decision support for β-thalassemia invasive prenatal diagnosisPLOS ONE

Dear Dr. Chen,

Thank you for submitting your manuscript to PLOS ONE. After careful consideration, we feel that it has merit but does not fully meet PLOS ONE’s publication criteria as it currently stands. Therefore, we invite you to submit a revised version of the manuscript that addresses the points raised during the review process.

We look forward to receiving your revised manuscript.

Kind regards,

J Francis Borgio, Ph.D.,

Academic Editor

PLOS ONE

“This study was funded by the Non-profit Central Research Institute Fund of Chinese Academy of Medical Sciences (2019PT310012) and National Nature Science Foundation of China (81960574, 82060578).”

Reviewers' comments:

Reviewer's Responses to Questions

**Comments to the Author**

1. Is the manuscript technically sound, and do the data support the conclusions?

Reviewer #1: Yes

Reviewer #2: Yes

2. Has the statistical analysis been performed appropriately and rigorously? 

Reviewer #1: Yes

Reviewer #2: Yes

3. Have the authors made all data underlying the findings in their manuscript fully available?

Reviewer #1: Yes

Reviewer #2: Yes

4. Is the manuscript presented in an intelligible fashion and written in standard English?

Reviewer #1: Yes

Reviewer #2: Yes

5. Review Comments to the Author

Reviewer #1: Review Comments to the Author:

The authors tried to investigate the utilization of non-invasive approach to select the healthy or carrier fetus before invasive procedures by using an alternative method, pseudo-tetraploid genotyping in concordance with gold standard invasive method and provided a valuable novel insight into patient stratification management on β-thalassemia prenatal diagnosis. Although much is known about the noninvasive approaches from various publications, they tried to add novel information to the literature by performing this study in chines’ population. The study design, materials and methods and the statics used are appropriate however there are some comments to be addressed, which are noted below.

Major comments:

1. First, much is known about this subject and there are numerous publications discussing this issue. In this paper, it is not very clear in which ways they contributed "novel" to the literature in correlation with other noninvasive approaches using maternal cell free DNA. This must be emphasized clearly and discussed accordingly.

2. The authors should clarify what are the fetal markers used to confirm the presence of fetal cell-free DNA in maternal plasma/blood.

3. Clinical data of the patients and the diversity of the mutations is limited. I suggest broadening the clinical data of the patients and discuss other types of mutations such as deletions, insertions and how these mutations could be applied to this technique. Additionally, it would be useful to discuss the concordant and discordant results.

4. Are the results confirmed by another testing method?

5. The "Discussion" is rather short. It would be better to expand it regarding the clinical data of the patients together with beta thalassemia mutation spectrum in different populations, the molecular techniques used in invasive and non-invasive testing with the up-to-date literature.

6. “Discussion” section is lack of references, it very important to discuss the results of current study in agreement or disagreement with other studies.

Minor comments:

1. There are some grammatical errors, and it should be revised throughout the manuscript where appropriate.

2. It is better to mention the objectives of the study and type of study in the abstract section.

3. It would be better to add more recent and up to date publication and delete old ones unless they are crucial.

4. Mention the name and version of software used for statistical purposes.

5. Also please explain the abbreviations used in the figure and give legend of the figure.

Reviewer #2: The manuscript describes a non invasive methodology to be used a tool for the stratification of patients before invasive procedures on β-thalassaemia prenatal diagnosis.

The authors used previously developed and evaluated methodology, the pseudo-tatraploid genotyping coupled with optimized bioinformatics algorithms to improve sensitivity and specificity. Their results were successful in predicting and linking the cfDNA haplotype to the beta allele. They presented here a very promising and intelligent predictive tool for fetal haplotyping using cell free fetal DNA in the high background of maternal DNA.

This predictive tool can be especially beneficial for the pregnancies where the father is not present or the genotype is unknown. However, in cases where the parents are both available, it would be more efficient, less costly and more specific to test directly the parents using genomic DNA, in order to predict the risk of the pregnancy having a beta thalassaemia child. So I recommend that screening the couple before hand could avoid having to apply this tool which is more complicated and more costly.

6. PLOS authors have the option to publish the peer review history of their article (what does this mean?). If published, this will include your full peer review and any attached files.

Reviewer #1: No

Reviewer #2: No

---

## [Author Response · Author response to Decision Letter 0]

17 Jan 2023

Dear Editor,

Our revised manuscript, "Integration of targeted sequencing and pseudo-tetraploid genotyping into clinical assisted decision support for β-thalassemia invasive prenatal diagnosis", has been resubmitted online to PLOS ONE.

We revised the manuscript according to the reviewers' comments. In the revised manuscript, we improved the writing by correcting typos, grammar, and phrase errors. Detailed presentations of patient clinical data, validation methods, and statistical methods are supplemented. Additionally, the discussion section has been comprehensively revised, adding descriptions of the spectrum of β-thalassemia mutations in different populations, molecular techniques used in invasive and noninvasive testing, and the latest literature.

Following the journal requirements, the manuscript has been restructured to meet the journal style requirements accordingly, including but not limited to text conversion of subsection titles, reformatting the tables, and updating the literature citation format, as indicated in the manuscript files. In addition, this study was funded by the Non-profit Central Research Institute Fund of Chinese Academy of Medical Sciences (2019PT310012) and National Nature Science Foundation of China (81960574, 82060578). The funders had no role in study design, data collection and analysis, decision to publish, or preparation of the manuscript.

Genetic data on thalassemia is prohibited from being transmitted to the public database temporarily by Chinese Law, but it can be disclosed in a restricted way. If necessary, please contact to Ethics Committee of the First Affiliated Hospital of Guangxi Medical University (Tel: +867715356557, Fax: +867715359801, Address: No.6, Shuangyong Road, Qingxiu District, Nanning, Guangxi province, 530021, China).

Thank you very much for your time handling our manuscript and the valuable suggestions given by the reviewers. We sincerely hope the revised manuscript could meet the merits and will be finally accepted for publication.

To reviewers

These comments are precious and helpful for revising and improving the manuscript. I have read through the comments carefully and revised the manuscript based on these comments. We revised the manuscript with the final "Clean Version", and point-to-point responses are also presented as follows, precisely describing what amendments have been made to the manuscript and where these can be viewed.

Responses to reviewer 1

1) Comment: First, much is known about this subject and there are numerous publications discussing this issue. In this paper, it is not very clear in which ways they contributed “novel“ to the literature in correlation with other noninvasive approaches using maternal cell-free DNA. This must be emphasized clearly and discussed accordingly.

My response: Thank you for your constructive comments. We have added a separate paragraph in the revised discussion section to give a detailed description instead of the previous unclear statement. Compared with the relative haplotype dosage (RHDO) and relative mutation dosage (RMD) algorithmic models, the "integrated target sequencing and pseudo-tetraploid genotyping" strategy has the advantage of detecting multiple sites in multiple patients at a single time. In addition, the pseudo-tetraploid algorithm can perform noninvasive prenatal diagnosis of the fetus and construct the fetal genotype without knowing the proband's genotype or obtaining a large amount of parental DNA information. (Page18-19, Line310-329)

2) Comment: The authors should clarify what are the fetal markers used to confirm the presence of fetal cell-free DNA in maternal plasma/blood.

My response: I am sorry for the unclear presentation. We used information on population linkage disequilibrium at additional adjacent loci upstream and downstream of the target locus, the difference between fetal and maternal genetic linkage, to assess fetal concentration and ensure that it was above the lower detection limit. A description of this section was added in the “Design principles“ section in the revised manuscript. ( Page8, Line159-166)

3) Comment: Clinical data of the patients and the diversity of the mutations is limited. I suggest broadening the clinical data of the patients and discuss other types of mutations such as deletions, insertions and how these mutations could be applied to this technique. Additionally, it would be useful to discuss the concordant and discordant results.

My response: Thank you very much for your suggestion. The clinical data of the enrolled patients were supplemented in Table 1, including pregnancy age, gestation week, weight, height, body mass index, reproductive history, and family history. Theoretically, this method is also suitable for detecting deletion and insertion mutations, but further large-scale studies are needed to confirm this. A total of 46 β-thalassemia mutations have been reported in the Chinese populations. Eight types, including HBB:c.126_129delCTTT, HBB: c.316-197C>T, HBB: c.216_217insA, HBB: c.52A>T, HBB: c.-78A>G, HBB: c.79G>A, HBB: c.-79A>G, HBB: c.130G>T accounted for more than 95% of the total number of β-thalassemia. On this basis, considering the economic factors and ethnic specificity of mutations, we designed only a test containing 11 common mutations to meet the epidemiological and clinical needs. This part was also supplemented in the discussion. (Page16-17, Line272-285) In addition, the detection rate and accuracy of the seven genotypes of cfDNA in maternal peripheral blood are variable, leading to the consistency and inconsistency of the test results. A more detailed discussion has been added to the revised manuscript. (Page19, Line338-357)

4) Comment: Are the results confirmed by another testing method?

My response: I am sorry for the unclear presentation. All fetal genotypes were subjected to standard invasive prenatal diagnosis according to standard protocols for molecular diagnostic testing [29, 30] at the Guangxi Key Thalassemia Research Laboratory. Chorionic villus or amniotic fluid samples of the fetuses were collected by ultrasound-guided transabdominal chorionic villus sampling or amniocentesis, and then DNA was extracted for PCR. Following that, reverse dot blot hybridization was performed, with 24 oligonucleotide probes arranged on the hybridization membrane to detect seven common and ten rare sites of β-thalassemia mutations in the Chinese population. (Page7, Line134-140)

5) Comment: The “Discussion“ is rather short. It would be better to expand it regarding the clinical data of the patients together with beta-thalassemia mutation spectrum in different populations, the molecular techniques used in invasive and non-invasive testing with the up-to-date literature.

My response: Thank you very much for your constructive suggestions. The “Discussion“ section has been revised accordingly. (Page15-21, Line255-368)

6) Comment: “Discussion“ section is lack of references, it very important to discuss the results of current study in agreement or disagreement with other studies.

My response: The entire “Discussion“ section has been revised accordingly, adding nearly 1300 words for a detailed description. (Page15-21, Line255-368)

7) Comment: There are some grammatical errors, and it should be revised throughout the manuscript where appropriate.

My response: I have invited an English speaker to polish the manuscript, and “Track Changes“was used to indicate the modified place in the revised “Track Changes Version” manuscript.

8 Comment: It is better to mention the objectives of the study and type of study in the abstract section.

My response: Thank you very much for your suggestion. The "Abstract" section has been modified accordingly. (Page2, Line26-29)

9) Comment: It would be better to add more recent and up-to-date publication and delete old ones unless they are crucial.

My response: As suggested by the reviews, we added the more recent and up-to-date publications and removed old ones. (Page22-31, Line392-591)

10) Comment: Mention the name and version of the software used for statistical purposes.

My response: The name and version number of the statistical software have been added to the "Statistical analysis" chapter in the revised article. (Page8, Line172-173)

11) Comment: Also please explain the abbreviations used in the figure and give legend of the figure.

My response: I am sorry for the unclear presentation. The figure title and legend have been rewritten. All abbreviations used in the figure have been explained, and the pictures have been more clearly illustrated. (Page11, Line209-217)

Responses to reviewer 2

1) Comment: The manuscript describes a non-invasive methodology to be used a tool for the stratification of patients before invasive procedures on β-thalassaemia prenatal diagnosis.

The authors used previously developed and evaluated methodology, the pseudo-tatraploid genotyping coupled with optimized bioinformatics algorithms to improve sensitivity and specificity. Their results were successful in predicting and linking the cfDNA haplotype to the beta allele. They presented here a very promising and intelligent predictive tool for fetal haplotyping using cell free fetal DNA in the high background of maternal DNA.

This predictive tool can be especially beneficial for the pregnancies where the father is not present or the genotype is unknown. However, in cases where the parents are both available, it would be more efficient, less costly and more specific to test directly the parents using genomic DNA, in order to predict the risk of the pregnancy having a beta thalassemia child. So I recommend that screening the couple beforehand could avoid having to apply this tool which is more complicated and more costly.

My response: Thank you for the helpful comment. To date, all noninvasive prenatal diagnostic strategies for thalassemia have failed to meet clinical needs compared with traditional invasive diagnostic methods. The main limiting factors are specificity and sensitivity, complex bioinformatics methods, and economic cost. Admittedly, prenatal diagnosis is the only effective way to avoid the birth of the affected fetus. If both parents are present, direct testing of the parents using genomic DNA to predict the risk of a pregnancy giving birth to a child with β-thalassemia would be a more effective, cost-effective, and specific approach. Our strategy is intended only as a reference for clinicians and not as a replacement for invasive testing. Our implementation aims to explore and refine this strategy with a view to its application in clinical practice soon. After all, safe and effective noninvasive prenatal diagnostic techniques have always been the direction and goal of prenatal diagnosis of genetic diseases such as thalassemia. The description concerning this comment was re-written between Pages 20, Lines 358-368, in the "Discussion" section.

---

## [Decision Letter · Decision Letter 1]

14 Mar 2023

Integration of targeted sequencing and pseudo-tetraploid genotyping into clinically assisted decision support for β-thalassemia invasive prenatal diagnosis

PONE-D-22-16457R1

Dear Dr. Chen,

We’re pleased to inform you that your manuscript has been judged scientifically suitable for publication and will be formally accepted for publication once it meets all outstanding technical requirements.

Kind regards,

J Francis Borgio, Ph.D.,

Academic Editor

PLOS ONE

Additional Editor Comments (optional):

Revised MS can be accepted

Reviewers' comments:

Reviewer's Responses to Questions

**Comments to the Author**

1. If the authors have adequately addressed your comments raised in a previous round of review and you feel that this manuscript is now acceptable for publication, you may indicate that here to bypass the “Comments to the Author” section, enter your conflict of interest statement in the “Confidential to Editor” section, and submit your "Accept" recommendation.

Reviewer #1: All comments have been addressed

Reviewer #3: All comments have been addressed

2. Is the manuscript technically sound, and do the data support the conclusions?

Reviewer #1: Yes

Reviewer #3: Yes

3. Has the statistical analysis been performed appropriately and rigorously? 

Reviewer #1: Yes

Reviewer #3: Yes

4. Have the authors made all data underlying the findings in their manuscript fully available?

Reviewer #1: Yes

Reviewer #3: Yes

5. Is the manuscript presented in an intelligible fashion and written in standard English?

Reviewer #1: Yes

Reviewer #3: Yes

6. Review Comments to the Author

Reviewer #1: No concerns regarding publication data, ethics etc. Author has addressed the comments and incorporated them into the document.

Reviewer #3: (No Response)

7. PLOS authors have the option to publish the peer review history of their article (what does this mean?). If published, this will include your full peer review and any attached files.

Reviewer #1: No

Reviewer #3: No

---

## [Editor Report · Acceptance letter]

27 Mar 2023

PONE-D-22-16457R1 

Integration of targeted sequencing and pseudo-tetraploid genotyping into clinically assisted decision support for β-thalassemia invasive prenatal diagnosis 

Dear Dr. Chen:

I'm pleased to inform you that your manuscript has been deemed suitable for publication in PLOS ONE. Congratulations! Your manuscript is now with our production department. 

Kind regards, 

on behalf of

Dr. J Francis Borgio 

Academic Editor

PLOS ONE